# Factors Influencing Sexual Violence Situation Witnessing Experience: For Korean Occupational Soldiers

**DOI:** 10.3390/bs13020090

**Published:** 2023-01-22

**Authors:** Hyekyung Kang

**Affiliations:** Department of Nursing, Joongbu University, Chubu-myeon, Geumsan-gun 32713, Chungcheongnam-do, Republic of Korea; kanghk@joongbu.ac.kr

**Keywords:** sexual violence awareness, gender consciousness, experience of witnessing sexual violence, social self-efficacy, self-esteem

## Abstract

This study explored the influencing factors related to the possibility of a military colleague’s intervention by comparing the awareness of sexual violence, gender consciousness, social self-efficacy, and self-esteem of Korean occupational soldiers related to the experience of witnessing sexual violence. From 4 September to 3 November 2022, an online survey was conducted targeting occupational soldiers working in all regions of South Korea by collecting snowball samples with gender allocation applied. Subjects responded to questions about sexual violence awareness, gender consciousness, witnessing sexual violence, social self-efficacy, and self-esteem. Data were collected until the number of professional soldiers with witnessing experience met the minimum target number by checking whether they had witnessed sexual violence among the response results. The collected data were analyzed using descriptive statistics, Pearson’s correlation analysis, and logistic regression. There were 102 subjects (40.9%) who had witnessed sexual violence by their military colleagues, and there were significant differences in sexual violence awareness, gender awareness, and social self-efficacy according to whether they had witnessed sexual violence. Compared to the unexperienced group, the group who witnessed sexual violence by their military colleagues showed significantly higher sexual violence awareness by 2.01 times and social self-efficacy by 2.27 times. In order to prevent sexual violence among occupational soldiers, it is necessary for military colleagues to play the role of active bystanders and witnesses. To this end, it is necessary to develop an educational direction and bystander’s witnessing program related to the improvement of the unit’s sexual violence awareness, gender awareness, and social role as a soldier.

## 1. Introduction

In Korea, as sexual violence deaths are constantly occurring among soldiers at a level that threatens the atmosphere of the organization, social interest in it has recently increased. Sexual violence refers to all sexual acts that cause mental and physical damage to others, including sexual harassment, sexual abuse, and sexual assault [1]. Moreover, many victims of sexual violence suffer from post-traumatic stress for a long time following the incident, due to anxiety and depression, regardless of the type of sexual violence involved [2,3].

However, the military insists on not revealing most of the incidents and accidents that occur because of the specific organization’s fraud and security concerns, and the atmosphere is one of quietly covering up the case for the unit involved, with the view that the victim is responsible, to at least some extent. As a result, sexual violence occurs frequently in the military. There were 771 cases of military sexual violence reported from 2020 to 2021. Additionally, regarding occupational soldiers, most of the perpetrators were male superiors, and most of the victims were women with less than five years of appointment [4,5,6]. The actual reporting rate of victims of military sexual violence is 32.7% [7,8], and considering the general male–female ratio of such cases, it is believed that the actual rate of occurrence is higher. This low reporting rate indicates that it is difficult for victims to receive an active response from the military. This is because these cases usually end with superiors offering conciliation under the pretext that it may lower the military organization’s morale, and if such cases are actively handled by the victim, there are fears of tangible and intangible retaliation in the form of bullying and omission from promotion, and loss of reputation within the organization [7,9].

However, if proper actions are not taken to respond appropriately to sexual violence cases within the military, it might lead to various socially negative consequences involving the deterioration of individual health and the quality of life in terms of personal aspects [10] and the danger to national security in terms of military aspects [11].

To properly cope with and prevent sexual violence in the military, the community role of military colleagues who share the military’s specificity and environment is more important than a victim-centered approach. To be specific, the military, which maintains communication and interpersonal relationships by class, is more closed, hierarchical, and male-oriented than any other workplace, so the problem cannot be solved just by victims of sexual violence reporting the incidents. Moreover, the military—in which perpetrators are usually superiors with the authority to decide training procedures, missions, promotions, and unit movements [4,5]—could lose its combat power, the key aspect of the military, as the morale of the victim and the military unit could be affected if military colleagues who train and live together for a long time do not work together to solve the issue.

Therefore, when the unit is at risk of sexual violence, its military colleagues can become bystanders and witnesses. If a military colleague who is aware of the situation plays the role of an active bystander, he or she can become not just a witness but also an unofficial social controller for risk behavior such as sexual violence, reducing incidents involving sexual violence [12,13]. Therefore, the prevention of sexual violence in the military requires a preventive strategy for peer-centered intervention, which follows the victim-centered approach, and includes post-incident protection [14,15,16]. Accordingly, to play an active role as a bystander, it is necessary to be adequately aware of the risk of sexual violence and possess the gender consciousness and right awareness of sexual violence that match the trend of the times and social values [17]. In addition, it is also possible to perform this role with a sense of efficacy as a member of an organization of occupational soldiers and with inner positivity and respect for oneself living in the present. In other words, self-efficacy and self-esteem to perceive one’s values positively—by being aware of the sexual violence risk of military colleagues working together in the unit, having sexual awareness and being aware of sexual violence, and having expectations of the ability to help military colleagues in times of difficulty—can affect the role of active bystanders in the event of sexual violence [16,18].

Therefore, this study aimed to investigate the influencing factors related to the possibility of peer intervention by comparing the usual perception of sexual violence, gender consciousness, self-esteem, and the social self-efficacy of Korean occupational soldiers according to the witnessing of sexual violence or lack thereof. This can be used as basic data to design educational directions and policies regarding the role of military colleagues in the future as active bystanders for the prevention of sexual violence.

## 2. Methods

### 2.1. Study Design

This study was designed to compare sexual violence awareness, gender awareness, social self-efficacy, and self-esteem between Korean occupational soldiers who have witnessed sexual violence and those who have not, and to investigate influencing factors related to the involvement of military colleagues.

### 2.2. Participants

Data collection was conducted through snowball sampling with gender-equal assignment by delivering the address of an online survey platform from 4 September to 3 November 2022 to subjects who were interested in this study and expressed their intention to participate. The survey was conducted until data from at least 233 people were received. The process was designed so that the survey began when the subjects accessed the survey platform, read the explanation for the study, and chose to agree to participate. The G*power 3.1.2 program was used to calculate the number of subjects, and the minimum target number was at least 210 people in total, with 105 people in each group. This was calculated when the independent sample *t*-test analysis method, bilateral test, median effect size 0.50, significance level 0.05, and power 0.95 were set. In this study, data were collected until the number of occupational soldiers with witness experience met the minimum target of 110 surveys to compare the characteristics of each variable between groups according to the presence or absence of sexual violence witness experience. Therefore, the final number of study subjects totaled 249 respondents. This number excluded 8 respondents out of 110 who witnessed sexual violence and 13 respondents out of 160 who did not witness sexual violence.

### 2.3. Measures

#### 2.3.1. Participants’ Demographic Characteristics

Demographic measures included age, gender, marital status, living arrangements, education level, economic status, and prior sexual violence education in the military.

#### 2.3.2. Gender Consciousness

To measure the gender consciousness, The Korean Gender Egalitarianism Scale-Short Form was used, which was developed by the Korea Women’s Policy Institute [19]. This scale was validated in 2016 and 2018. There were a total of 12 items, all items on a 4-point Likert scale (1 = strongly disagree, 4 = strongly agree), with a higher score mean lower awareness of gender equality. Cronbach’s α was 0.852 in this study.

#### 2.3.3. Sexual Violence Awareness

For sexual violence awareness, the study used the 15-item, 4-point (1 = strongly disagree, 4 = strongly agree) version of the Ministry of Gender Equality & Family (2019) questionnaire [3]. This tool refers to the degree of recognition of sexual violence in relation to sexual violence and is conducted every three years for male and female adults aged 19 or older in South Korea. Five items for dating violence and cyber sexual assault questions were added to the 10 questionnaires on sexual violence awareness in 2016. The higher the score, the lower the awareness of sexual violence. In this study, Cronbach’s α = 0.87.

#### 2.3.4. Experience of Witnessing Sexual Violence

This tool asks about witnessing experiences of sexual violence in the unit, and it was used by the National Human Rights Commission of Korea in 2019 to investigate the human rights situation in the military [8]. Among them, six questions were used about the experiences of witnessing dangerous behaviors in the unit—such as verbal violence, bullying and sexual harassment—and measures taken after witnessing them. Among them, those who responded to the five questions about the degree of involvement in witnessing were classified as experience of witnessing, and those who answered “no witnesses” were classified as nonexperience of witnessing.

#### 2.3.5. Social Self-Efficacy

Self-efficacy refers to a belief in one’s own ability to succeed in certain situations, and it is a reflection of confidence in the ability to exert control efficiently to handle special situations [20]. For social self-efficacy in this study, the tool of Kang and Kim (2013) [21] was used. This tool measures the positive self of helping others when they see a colleague in need and their estimation of their ability to help. There are a total of eight questions and a 6-point Likert scale; the higher the score, the higher the social self-efficacy. At the time of development, Cronbach’s α was 0.84, and in this study, it was 0.89.

#### 2.3.6. Self-Esteem

Self-esteem was measured using the Rosenberg Self-Esteem Scale (1965) [22]. This scale has good psychometric properties. The scale contains 10 items and the score of each item ranges from 1 to 4 points, with a higher score indicating higher self-esteem. Here, Cronbach’s α was 0.831.

### 2.4. Data Analysis

The collected data were analyzed using the Statistical Package for the Social Sciences (SPSS) 23.0/PC software (IBM Corp., Armonk, NY, USA). The χ^2^ test was used to compare the general characteristics between the two groups according to the number and percentage of subjects and the presence or absence of witness experience of sexual violence. The difference in sexual violence-related perception, gender consciousness, self-esteem, and social self-efficacy between the two groups was analyzed using the *t*-test, the difference in variables according to the experience of witnessing sexual violence and general characteristics was analyzed using the *t*-test and one-way ANOVA, and the correlation between variables was analyzed using the Pearson correlation coefficient. The cross-ratio according to the experience of witnessing sexual violence by group based on the average value of major variables was analyzed using logistic regression based on the set reference group.

## 3. Results

### 3.1. Differences According to Demographic Characteristics and Experience of Witnessing Sexual Violence

The subjects in this study comprised 102 women (41%) and 147 men (59%). Of these subjects, 225 individuals (90.4%) had received sexual violence prevention education in the military, and 102 individuals (40.9%) had witnessed sexual violence perpetrated by military colleagues. There was a significant difference in sexual violence consciousness (*p* < 0.001), gender consciousness (*p* < 0.01), and social self-efficacy (*p* < 0.05) between the group that had experienced sexual violence situations and the group that had not (Table 1).

### 3.2. Correlation between Awareness of Sexual Violence, Gender Consciousness, Social Self-Efficacy, and Self-Esteem According to the Experience of Witnessing Sexual Violence Situations

In the case of the group that had witnessed sexual violence situations, the relationships between sexual violence awareness and gender consciousness (*r* = 0.82, *p* < 0.001), sexual violence awareness and self-esteem (*r* = 0.16 and *p* < 0.05), gender consciousness and social self-efficacy (*r* = 0.14, *p* < 0.05), and social self-efficacy and self-esteem (*r* = 0.51, *p* < 0.01) were found to be statistically significant. In the case of the group comprising individuals who had not witnessed sexual violence situations, the relationships between sexual violence awareness and gender consciousness (*r* = 0.52, *p* < 0.001), gender consciousness and social self-efficacy *(r* = 0.56, *p* < 0.001), and social self-efficacy and self-esteem (*r* = 0.56, *p* < 0.001) were found to be statistically significant (Table 2).

### 3.3. Cross-Ratio of Sexual Violence Awareness, Gender Consciousness, Social Self-Efficacy, and Self-Esteem According to the Experience of Witnessing Sexual Violence Situations

Logistic regression analysis was conducted to determine the effect of sexual violence awareness, gender consciousness, social self-efficacy, and self-esteem on the experience of witnessing sexual violence. The results showed that sexual violence awareness was 2.01 times greater (95% CI = 1.08–2.13) and social self-efficacy was 2.27 times greater (95% CI = 0.67–3.10) in the group with witness experience, and both values were statistically significant. However, there was no statistical difference in gender consciousness and self-esteem (Table 3).

## 4. Discussion

This study was conducted among Korean occupational soldiers in order to prevent the occurrence of sexual violence risk behavior, and it attempted to identify the degree of sexual violence awareness, gender consciousness, self-esteem, and social self-efficacy according to experiences of witnessing sexual violence situations among military colleagues. It also attempted to identify factors affecting sexual violence-related awareness.

The results of this study showed a significant difference in sexual violence-related perception between the experienced and non-experienced groups of the experience of witnessing sexual violence situations; the higher the related perception, the higher (about twice) the experience of witnessing sexual violence situations. The intervention of military colleagues within the military unit where sexual violence occurred starts with where sexual violence occurs; the intervention of a military colleague starts with the witness experience of a military colleague who interprets the situation as a dangerous act [23]. In order for bystanders to intervene in a particular event, they should recognize the situation, interpret it as dangerous, feel the responsibility to intervene, decide on the appropriate action, and then undertake a decision-making process [24]. Therefore, an intervention would not occur in situations where bystanders do not recognize the situation itself, do not feel personal responsibility, do not have the ability to intervene, or do not intend to intervene. The presence or absence of witness experience of sexual violence situations may affect the degree of perception of the sexual violence situation [23], so improvement of related awareness education in the military unit is required. The military’s sexual violence prevention education and bystander intervention programs have raised sexual awareness and heightened the willingness to intervene [25,26], supporting the notion that education, such as bystander intervention training, is a worthwhile investment for the military organization.

In addition, the results of this study showed a relationship between sexual violence awareness and gender consciousness, regardless of whether sexual violence was witnessed. Although the number of female soldiers is increasing, most military members are still male, and given the reality that sexual violence in the Korean military occurs between male perpetrators and female victims, it can be assumed that the degree of gender consciousness in military organizations is related to sexual violence occurrence and witness intervention. This supports the context of a previous study in which the witness’ gender and attitude toward sexual violence were related to predicting the possibility of intervention in a sexual violence situation [27]. In workplaces with a large number of men, there are often cases in which sympathy lies with the male perpetrator, owing to the same gender identity. In fact, in some cases, a male colleague may disregard the situation or fail to intervene even after witnessing a sexual violence situation because of the fear that his help or concern for female victims could be considered a flaw in masculinity by other men [13,15,28]. Therefore, it is necessary to improve organizational culture in terms of gender consciousness, as the decision of peer intervention by the witness can be affected by the organization’s gender-related cultural perception. However, caution should be exercised against unfairly categorizing male soldiers as potential perpetrators or passive bystanders. Even in the same situation, men and women can perceive the degree of risk of sexual violence differently [29,30]. Therefore, it is necessary to explore obstacles to the intervention of active bystanders.

Helping others in social relationships is related to the recognition of values and social expectations. In particular, co-workers are persons who strive to achieve the goals of the organization in which they belong and, as a result, spend as much time together as they spend with their family, and sometimes more. Moreover, military colleagues protect each other’s lives, and live together and they work together for their country; military personnel have high recognition and expectations regarding social relations [31]. In this study, the social self-efficacy showed a significant difference between the experienced group and the non-experienced group of witnessing sexual violence situations; the higher the social self-efficacy, the higher the experience of witnessing the sexual violence situation (by about 2.2 times). This result supports previous studies which found that, as self-efficacy increased, the bystander’s intention to intervene increased [32].

Social self-efficacy indicates expectations for performance in social situations formed by interactions with others [20], and self-esteem, which means respecting oneself [22], extends to the attitude toward oneself felt in relationships with colleagues or other important persons [33]. Prior research found that most of those who intervened in sexual violence situations felt positive about their actions, whereas most people who did not intervene reported negative feelings about their inaction [34]. This suggests that helping others can enhance self-esteem and social self-efficacy, which is in line with the results of this study. Therefore, the intervention of military colleagues in dangerous situations, such as those involving sexual violence, is an act of helping others, and it can contribute to the achievement of the organization’s ultimate goals by strengthening the social relationship of military unit members.

## 5. Conclusions and Suggestions

This study aimed to identify the influencing factors related to the possibility of intervention by military colleagues according to experience with witnessing sexual violence situations. Active resolution related to helping military colleagues in the event of witnessing sexual violence is good for individuals, and even if they do not solve the problem, acting with interest is a good attempt to prevent sexual violence. This can ultimately affect the organizational culture of the military and further improve the social self-efficacy and self-esteem of members. Furthermore, to prevent sexual violence among occupational soldiers, military colleagues need to intervene as bystanders and witnesses, and this requires the development of educational directions and witness programs to improve the military unit’s awareness of sexual violence, gender consciousness, and the social role as soldiers.

As this study involved only Korean occupational soldiers, there are limitations in generalizing the research results. In particular, as witnessing and intervening in dangerous behaviors such as sexual violence can be affected by the characteristics of each country and its military and the degree of danger in the working environment; therefore, follow-up studies on sexual violence-related perceptions according to military characteristics and military environments are suggested. The social self-efficacy tool that was used in this study was developed for college students, for whom the stakes are not as high. So social self-efficacy related to help behavior might not be effectively reflected in the hierarchical military. Therefore, it is necessary to explore tools related to intervening with and overcoming dangerous situations in the future. In addition, this study was surveyed during the COVID-19 pandemic. As the COVID-19 pandemic is potentially related to various psychological factors—such as depression, relationship withdrawal, stress, and interpersonal difficulties—factors that affect the subject’s social and psychological measurement results, including the number of sightings, need to be considered.

## Figures and Tables

**Table 1 behavsci-13-00090-t001:** Experience of witnessing sexual violence in general characteristics and main variables (N = 249).

Variables	Categories	Total (N = 249)	Experience of Witnessing-Yes (n = 102)	Experience of Witnessing-No (n = 147)	*t* or *x*^2^
		n (%) or M ± SD	n (%) or M ± SD	n (%) or M ± SD	
Gender	Female	102 (41.0)	35 (34.3)	67 (45.6)	2.71
	Male	147 (59.0)	67 (65.7)	80 (54.4)	
Age (year)	≦29	67 (26.9)	28 (27.5)	39 (26.5)	20.12
	30–39	79 (31.7)	34 (33.3)	45 (30.6)	
	40–49	60 (24.1)	22 (21.6)	38 (25.9)	
	≧50	43 (17.3)	18 (17.6)	25 (17.0)	
Marital status	Married	91 (36.5)	42 (41.2)	49 (33.3)	26.43
	Single	158 (63.5)	60 (58.8)	98 (66.7)	
Education level	≦High school	43 (17.3)	14 (13.7)	29 (19.7)	16.55
	College	77 (30.9)	31 (30.4)	46 (31.3)	
	University	101 (40.6)	45 (44.1)	56 (38.1)	
	≧Graduate school	28 (11.2)	12 (11.8)	16 (10.9)	
Living arrangements	Alone	104 (41.8)	42 (41.2)	62 (42.2)	19.32
	With family	113 (45.3)	47 (46.1)	66 (44.9)	
	With other	32 (12.9)	13 (12.7)	19 (12.9)	
Economic condition	High	33 (13.3)	15 (14.7)	18 (12.2)	1.12
	Middle	160 (64.2)	62 (60.8)	98 (66.7)	
	Low	56 (22.5)	25 (24.5)	31 (21.1)	
Education in the military	Yes	225 (90.4)	96 (94.1)	129 (87.8)	2.02
	No	24 (9.6)	6 (5.9)	18 (12.2)	
Sexual violence awareness		27.57 ± 6.48	22.59 ± 4.32	30.45 ± 7.05	7.29 *****
Gender consciousness		23.70 ± 6.26	20.31 ± 7.53	27.12 ± 4.72	8.18 ****
Social self-efficacy		27.88 ± 5.01	27.05 ± 8.78	27.90 ± 7.15	1.52 ***
Self-esteem		29.96 ± 7.64	28.06 ± 6.70	32.22 ± 4.56	2.95

Note. * *p* < 0.05, ** *p* < 0.01, *** *p* < 0.001.

**Table 2 behavsci-13-00090-t002:** Correlation among sexual violence awareness, gender consciousness, social self-efficacy, and self-esteem (N = 249).

Variables		1. Sexual Violence Awareness	2. Gender Consciousness	3. Social Self-Efficacy	4. Self-Esteem
		*r* (*p)*			
Experience of witnessing-Yes (n = 102)	1	1			
	2	0.82 (<0.001)	1		
	3	0.03 (0.482)	0.14 (<0.05)	1	
	4	−0.16 (<0.05)	0.25 (0.024)	0.51 (<0.01)	1
Experience of witnessing-No (n = 147)	1	1			
	2	0.52 (<0.001)	1		
	3	0.04 (0.478)	0.11 (<0.001)	1	
	4	0.08 (0.144)	0.13 (0.014)	0.56 (<0.001)	1

**Table 3 behavsci-13-00090-t003:** Odds ratio on sexual violence awareness, gender consciousness, social self-efficacy, and self-esteem according to experience of witnessing sexual violence (N = 249).

Variables	Odds Ratio	95% CI *	*p*
Sexual violence awareness	2.01	1.08–2.13	<0.001
Gender consciousness	1.74	0.14–2.59	0.321
Social self-efficacy	2.27	0.67–3.10	0.017
Self-esteem	0.91	0.55–2.86	0.091

* Confidence interval.

## Data Availability

Data sharing is not applicable to this article as no datasets were generated or analyzed during the current study.

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
