# Peer review of "Factors Influencing Sexual Violence Situation Witnessing Experience: For Korean Occupational Soldiers"

_behavsci, 2023, doi:10.3390/bs13020090_

Round 1
Reviewer 1 Report
Method: Study design
Page 3/13, 1st paragraph
The research design is a plan to answer the research question. The article does not state the research question very clearly and, as a result, the plan to answer it is not clear either. The statement, “This study describes sexual violence awareness…”, is not appropriate for a study design section, where one would think that the word “plan” or something similar would be essential.
The efficacy issue
The author notes that the violence often involves perpetrators of superior rank acting against victims of inferior position, usually with less than 5 yrs experience (p. 2/13), which suggests that the fear of retaliation is the main reason why the victims hesitate to report the incident or prosecute the perpetrator (p. 2/13). To solve this problem, the author suggests that relying on “the community role of military colleagues” might be a more practical solution. Nevertheless, the same fear of retaliation that plagues the victims might be a factor in the military community’s behavior. In other words, what would motivate a military bystander to intervene, if this is going to be at the expense of his or her own military career? The author states that the subjects’ self-efficacy has been measured sufficiently by the scale for rescue behaviors developed by the author (Kang and Kim, 2013), but that scale was developed for college students, for whom the stakes are not as high. In addition, more recent research has showed that the self-efficacy theory is very problematic and cannot be taken for granted (Forsyth & Carey, in press). Efficacy scores may be influenced by response bias, as research participants may respond in ways that reflect well of them but do not really intend to act accordingly in real-life situations (Bandura, 1997). It seems that especially in a military setting, protecting bystanders’ intervention and whistle blowers as well as cultivating self- and social- efficacy may be as important as raising awareness of sexual violence. The author must address this issue before the proposed intervention (which has merit) has any hope of success.
Bandura, A. (1977). Self-efficacy: Toward a unifying theory of behavioral change. Psychological Review, 84(2), 191-215.
Bandura, A. (1997). Self-Efficacy: The exercise of control. New York, NY: W. H. Freeman.
Forsyth, A. D., & Carey, M. P. (in press). Problems in the measurement of self-efficacy: Review, critique, and recommendations. Health Psychology.
Author Response
Dear
Please see the attachment.
Thank you for your kind review
Sincerely,
Author

Reviewer 2 Report
Please add the following sections:
1. Conceptualization- Social Self Efficacy...
2. Add more recent literature in a comparative manner- compare for example with the US...
3. Strengthen the discussion section- Results categorization...
4. Recommendations - clear list to decision-maker to ensure it does not occur in the future.
Author Response

(The authors gave the same response as above.)

Round 2
Reviewer 1 Report
General Comment
This version of the article is much improved and more focused. However, the language is still problematic and needs refining. I have offered some suggestions below. The word to be corrected is in red and my comments are in the square brackets that follow.
PAGE 3
Methods
Study design
This study was designed to compare awareness of sexual violence, gender awareness, social self-efficacy, and self-esteem in [delete “in” and replace with “between”] Korean occupational soldiers who have witnessed sexual violence and [add: those who] have not, and to investigate influencing factors related to the of [delete “of”] involvement by [of] military colleagues.
PAGE 3
The process was designed such [so] that the survey began when the subjects accessed the survey platform, read the explanation for the study, and chose to agree to participate according to their wishes.
PAGE 3
Gender consciousness
I used The Korean Gender Egalitarianism Scale-Short Form developed by the Korea Women’s Policy Institute.1. [Avoid using the first-person pronoun “I”; turn it into “we”. For example: “For the purpose of X, we used The Korean Gender Egalitarianism Scale-Short Form, which was developed by…]
PAGE 4
Self-efficacy refers to one's own belief of ability by recognizing the possibility of success in certain situations and is a reflects confidence in the ability to exert control efficiently one can handle special situations. 20 In this study, self-efficacy means that self-efficacy has expanded socially to expectations of the ability to help colleagues in need and positive values for helping others.
[This sentence, which is crucial to the study, is very poorly written and makes very little sense. For one, you cannot define a term by using the same term in the definition, “self-efficacy means that self-efficacy has…” ]
PAGE 6
This supports the context of a previous study in which the witness' gender and attitude toward sexual violence are related to predicting the possibility [add: of] intervention in a sexual violence situation.
PAGE 6
However, caution should be exercised against unfairly category [categorizing] male soldiers as potential perpetrators or passive bystanders.
PAGE 6
Helping others in social relationships is related to the recognition of values and social expectations [add a period here and begin the next sentence with a capital letter, “In particular”] in particular, a co-worker at work is a person who achieves the goals of the organization I [avoid first person; replace with “one”] belong to together.
[Incorrect grammar that gets worse in the subsequent clauses. Perhaps replace with the following: “In particular, co-workers are persons who strive to achieve the goals of the organization in which they belong and, as a result, spend as much time together as they spend with their family, sometimes more.]
and in modern society, spends [example: “spend as much time together as they spend with their family, sometimes more.”] as much time as [add: with] my [avoid first person; replace with “one’s”] family, sometimes more than [add: with] my [avoid first person; replace with “one’s”] family.
PAGE 6
Prior research found that most of those who intervened in sexual violence situations feeling [felt] positive about their actions…
PAGE 7
The social self-efficacy tool [add: that] was used in this study have been [was] developed for college students, for whom the stakes are not as high.
Author Response
I appreciate your kind and detailed review.
I have revised my manuscript according to your corrections and advice.
Thank you again.
Sincerely,
Author